# Postprandial sleep mechanics in *Drosophila*

Keith R Murphy[1,2,3], Sonali A Deshpande[1], Maria E Yurgel[2], James P Quinn[1], Jennifer L Weissbach[1], Alex C Keene[2], Ken Dawson-Scully[2], Robert Huber[4,5], Seth M Tomchik[3], William W Ja[1,3]*

[1]Department of Metabolism and Aging, The Scripps Research Institute, Jupiter, United States; [2]Program in Integrative Biology and Neuroscience, Florida Atlantic University, Jupiter, United States; [3]Department of Neuroscience, The Scripps Research Institute, Jupiter, United States; [4]Radcliffe Institute for Advanced Study, Harvard University, Cambridge, United States; [5]JP Scott Center for Neuroscience, Mind and Behavior, Bowling Green State University, Bowling Green, United States

**Abstract** Food consumption is thought to induce sleepiness. However, little is known about how postprandial sleep is regulated. Here, we simultaneously measured sleep and food intake of individual flies and found a transient rise in sleep following meals. Depending on the amount consumed, the effect ranged from slightly arousing to strongly sleep inducing. Postprandial sleep was positively correlated with ingested volume, protein, and salt—but not sucrose—revealing meal property-specific regulation. Silencing of leucokinin receptor (Lkr) neurons specifically reduced sleep induced by protein consumption. Thermogenetic stimulation of leucokinin (Lk) neurons decreased whereas Lk downregulation by RNAi increased postprandial sleep, suggestive of an inhibitory connection in the Lk-Lkr circuit. We further identified a subset of non-leucokininergic cells proximal to Lkr neurons that rhythmically increased postprandial sleep when silenced, suggesting that these cells are cyclically gated inhibitory inputs to Lkr neurons. Together, these findings reveal the dynamic nature of postprandial sleep.

*For correspondence: wja@scripps.edu

**Competing interests:** The authors declare that no competing interests exist.

## Introduction

Despite accumulating evidence of interactions between sleep and metabolism, few studies have documented an increased propensity for sleep that animals might experience after meals. Results on human postprandial behavior have shown an increased sleep propensity following a meal, a lagging response, or no effect at all (*Orr et al., 1997*; *Zammit et al., 1995*; *Wells et al., 1998*; *Harnish et al., 1998*; *Stahl et al., 1983*). It has been proposed that postprandial sleep is not invariable; it may be absent in particular individuals or heavily dependent on specific food properties (*Stahl et al., 1983*). While the behavior likely exists, these intrinsic variabilities have limited our ability to study its molecular basis. A thorough examination of this behavior would be facilitated by use of an animal model although, at present, no clear model exists (*Watson, 2014*).

In *Drosophila* there is a well-documented sleep-metabolism interaction in which flies suppress sleep or increase locomotion when starved (*Lee and Park, 2004*; *Thimgan et al., 2010*; *Keene et al., 2010*). However, the acute effects of food consumption on sleep have not been tested, largely because there is no system available to do so. Here we establish a system for simultaneous measurement of discrete feeding events and sleep, allowing us to examine the mechanisms underlying their interaction.

**eLife digest** Many of us have experienced feelings of sleepiness after a large meal. However, there is little scientific evidence that this "food coma" effect is real. If it is, it may vary between individuals, or depend on the type of food consumed. This variability makes it difficult to study the causes of post-meal sleepiness.

Murphy et al. have now developed a system that can measure fruit fly sleep and feeding behavior at the same time. Recordings using this system reveal that after a meal, flies sleep more for a short period before returning to a normal state of wakefulness. The sleep period lasts around 20-40 minutes, with flies that ate more generally sleeping more.

Further investigation revealed that salty or protein-rich foods promote sleep, whereas sugary foods do not. By using genetic tools to turn on and off neurons in the fly brain, Murphy et al. identified a number of brain circuits that play a role in controlling post-meal sleepiness. Some of these respond specifically to the consumption of protein. Others are sensitive to the fruit fly's internal clock, reducing post-meal sleepiness only around dusk. Thus, post-meal sleepiness can be regulated in a number of different ways.

Future experiments are now needed to explore the genes and circuits that enable meal size and the protein or salt content of food to drive sleep. In nature, sleep is likely a vulnerable state for animals. Thus, another challenge will be to uncover why post-meal sleep is important. Does sleeping after a meal boost digestion? Or might it help animals to form memories about a food source, making it easier to find similar food in the future?

## Results

To simultaneously measure sleep and feeding of individual adult flies, we designed the Activity Recording CAFE (ARC), a system that couples machine vision tracking of capillary-based food consumption and animal motion (*Figure 1A–C*) (*Ja et al., 2007*; *Donelson et al., 2012*; *Deshpande et al., 2014*). Previous studies have shown that flies inactive for 5 min exhibit multiple hallmarks of sleep (*Shaw et al., 2000*; *Huber et al., 2004*; *Hendricks et al., 2000*). Using this standard, we extracted sleep parameters by defining sleep as periods of inactivity >5 min. Despite different housing and food consistency, we found sleep architecture to be similar to previous studies using high resolution object tracking, a method which has greater resolution of total sleep and sleep duration than the infrared-beam based alternative (*Figure 1C*, *Figure 1—figure supplement 1A–D*) (*Donelson et al., 2012*). Flies appeared to move less after meals than before—an effect more pronounced with larger meals—suggesting that sleep might also be greater after eating (*Figure 1D*). Animals also positioned themselves closer to the food following each meal (*Figure 1E–F*), consistent with observations that isolated flies sleep close to food (*Hendricks et al., 2000*) and the spatial tendency of flies driven to sleep by activation of neurons expressing short neuropeptide F, a satiety signaling hormone (*Shang et al., 2013*; *Lee et al., 2004*).

To examine the acute effects of feeding on sleep, we quantified the probability of sleep ($P_{sleep}$) relative to individual feeding events. The animals experienced a sharp decline in $P_{sleep}$ in the time approaching each meal, as expected since flies cannot sleep and feed simultaneously. Subsequent to the meal, $P_{sleep}$ rose rapidly and exceeded the respective pre-meal $P_{sleep}$ (*Figure 2A*). To quantify changes in sleep subsequent to each meal, we calculated the change in $P_{sleep}$ of respective time points before and after each event (i.e. $t^{+1} – t^{-1}$, $t^{+2} – t^{-2}$, . . .). This analysis revealed a rise in sleep lasting approximately 20–40 min with a maximum $\Delta P_{sleep}$ amplitude of 0.15 and 0.2 for the control strains, $w^{1118}$ and Canton-S, respectively (*Figure 2B*). The effect was observed in both males and females (*Figure 2—figure supplement 1A*). While the duration of sleep events prior to a meal is inherently limited by the time until an animal wakes to eat, sleep initiation is not limited and was more probable following a meal (*Figure 2—figure supplement 1B*).

In addition to periods of quiescence, increased arousal threshold is regularly used to demonstrate sleep in flies. To test if arousal threshold is increased after a meal, we adapted methodology from the *Drosophila* arousal tracking system in which animals are exposed to a series of increasing vibrational stimuli (0.8–3.2 $g$) once every hour in order to measure the intensity at which inactive animals

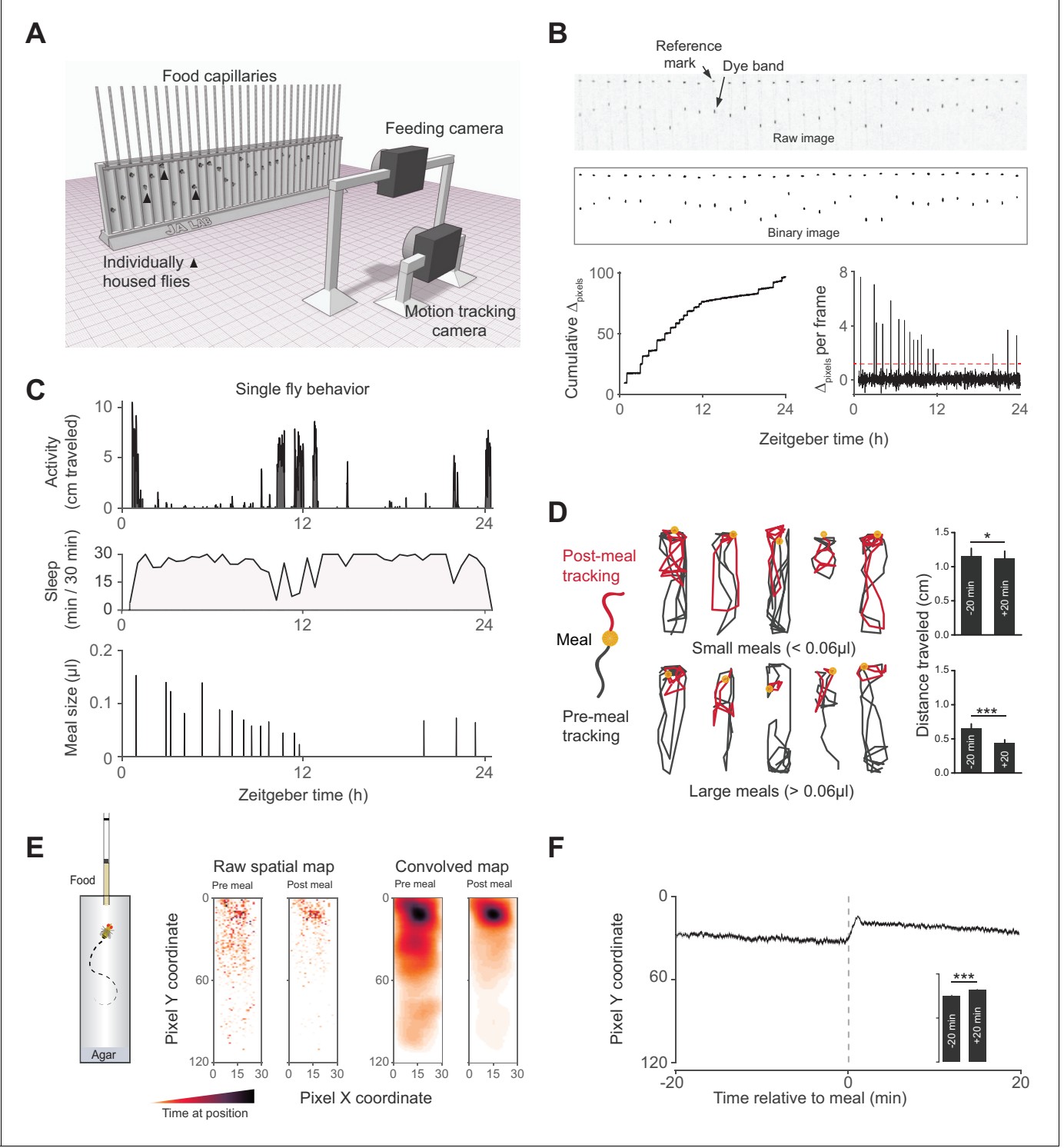

**Figure 1.** Activity Recording CAFE (ARC) facilitates simultaneous, high resolution measurements of food intake and motion in individual flies. (**A**) Schematic of ARC apparatus. Independent computer-controlled cameras record images of capillaries for feeding measurements and of flies for activity and sleep determination. (**B**) Raw image (one out of 1440 taken over 24 hr) of capillaries containing liquid food (top). Capillaries have an external reference mark and an internal dyed oil band overlaid the aqueous food. Images are thresholded to binary, from which cumulative pixel distances between the reference mark and dye band can be calculated from the image series (bottom). $\Delta_{pixels}$ per frame reveals individual feeding events, selected as events greater than 3.5 standard deviations above noise (red dashed line). (**C**) Activity, sleep (black shading) and feeding (μl) of an individual fly (activity and feeding in 1 min bins, sleep in 30 min bins). (**D**) Example motion traces of individuals before (black) and after (red) a meal (yellow circle). Motion traces and averages associated with small (<0.06 μl) or large (≥0.06 μl) meals are shown. n = 661 meals from 30 flies, $w^{1118}$; *p<0.05,
*Figure 1 continued on next page*

*Figure 1 continued*

***p<0.001, Wilcoxon matched-pairs sign rank test. (**E**) Raw and convolved spatial heat map showing time spent at each pixel coordinate in the 20 min before and after meals (291 feeding events from 15 *w1118* flies, Gaussian convolution). (**F**) Kymograph of average vertical position in time relative to meals (shaded line represents mean ± s.e.m.). The inset graph shows the average position over 20 min pre-meal (−20) and post-meal (+20) with the axis scaled to the parent graph. ***p<0.001, Wilcoxon matched-pairs sign rank test. All bars represent mean ± s.e.m.

The following figure supplement is available for figure 1:

**Figure supplement 1.** Influence of recording apparatus on sleep architecture.

respond (*Figure 2C*) (*van Alphen et al., 2013*; *Faville et al., 2015*). Consistent with previous studies, we found that arousal threshold initially increases with the amount of time that a fly is inactive, with the greatest change occurring in the first 5 min (*Figure 2D*) (*Shaw et al., 2000*; *Huber et al., 2004*; *van Alphen et al., 2013*; *Faville et al., 2015*). Since the maximum duration of inactivity is lower following a meal, irrespective of sleep probability (*Figure 2E*), we filtered arousal events to compare animals in similar stages of sleep. While animals inactive for more than 5 min are experiencing change in sleep depth, animals inactive for 0–5 min are transitioning into sleep. In this transition state animals showed a significantly greater arousal threshold in the 20, 40, and 60 min post-meal compared to animals during the respective pre-meal times (*Figure 2F*). Interestingly, the correlation between arousal threshold and postprandial sleep grew weaker when comparing animals in deeper stages of sleep (*Figure 2—figure supplement 2A–B*). This suggests that food intake transiently influences sleep induction rather than depth, which might be shaped solely by the amount of time that an animal is inactive.

We further considered the possibility that increases in immobility-derived sleep and arousal threshold could be artifacts of grooming behavior. To test this, we manually scored videos of animals during an ARC recording and annotated grooming events. Falsely calculated sleep, in which animals were grooming rather than immobile, accounted for ~7% of sleep surrounding meals and had statistically indistinguishable occurrence before and after a meal (*Figure 2G*, *Figure 2—figure supplement 3*). Re-analysis of sleep surrounding a meal, where sleep was defined as periods of immobility and a lack of grooming exceeding 5 min, lowered total sleep but did not alter the rise in postprandial sleep (*Figure 2H*).

We next asked how the quantity of food consumed might influence postprandial sleep. We plotted $P_{sleep}$ or $\Delta P_{sleep}$ for different consumption volumes and found that sleep following meals increased as a function of meal size, while pre-prandial sleep did not (*Figure 3A*, *Figure 3—figure supplement 1A*). Since average postprandial sleep lasted between 20–40 min, and since arousal threshold differences were most apparent in the first 20 min, we re-calculated $\Delta P_{sleep}$ for the 20 min surrounding each meal. We found a significant correlation between $\Delta P_{sleep}$ and consumption volume when meals were analyzed individually (*Figure 3—figure supplement 1B*) or binned (*Figure 3B*). Since this analysis used individual meals rather than individual animals, we tested whether animals that consumed meals more frequently could bias the observed effect. We performed a Monte Carlo simulation by iteratively recapitulating the analysis with an equal number of randomly selected meals from each animal and found that equal sampling accurately portrays the full data set (*Figure 3—figure supplement 2A–B*). Since sleep is derived from periods of inactivity, we considered whether increased consumption could affect mobility, which might in turn appear to be sleep. The change in movement rate after a meal ($\Delta_{speed}$) showed a much weaker correlation to meal size than $\Delta P_{sleep}$ (*Figure 3—figure supplement 2C*). Since speed is a strong indicator of mobility, we reasoned that immobility is not likely responsible for the sleep effect.

Although meal size has an apparent impact on postprandial sleep, the standard diet contains numerous physiological and nutritive components. To determine the potential contribution of individual components to postprandial sleep, we designed a paradigm that allowed us to test a range of feeding volumes or of ingested nutrients. To test volumetric effects, animals were fed a low protein concentration food (0.25% tryptone) to elicit feeding without impacting baseline sleep, since sucrose and sweet tasting substances have been shown to modulate sleep (*Keene et al., 2010*; *Linford et al., 2012*). We found a highly significant correlation between meal volume and 20 min $\Delta P_{sleep}$ (*Figure 4A*, *Figure 4—figure supplement 1A–B*). Larger meal volumes are also associated

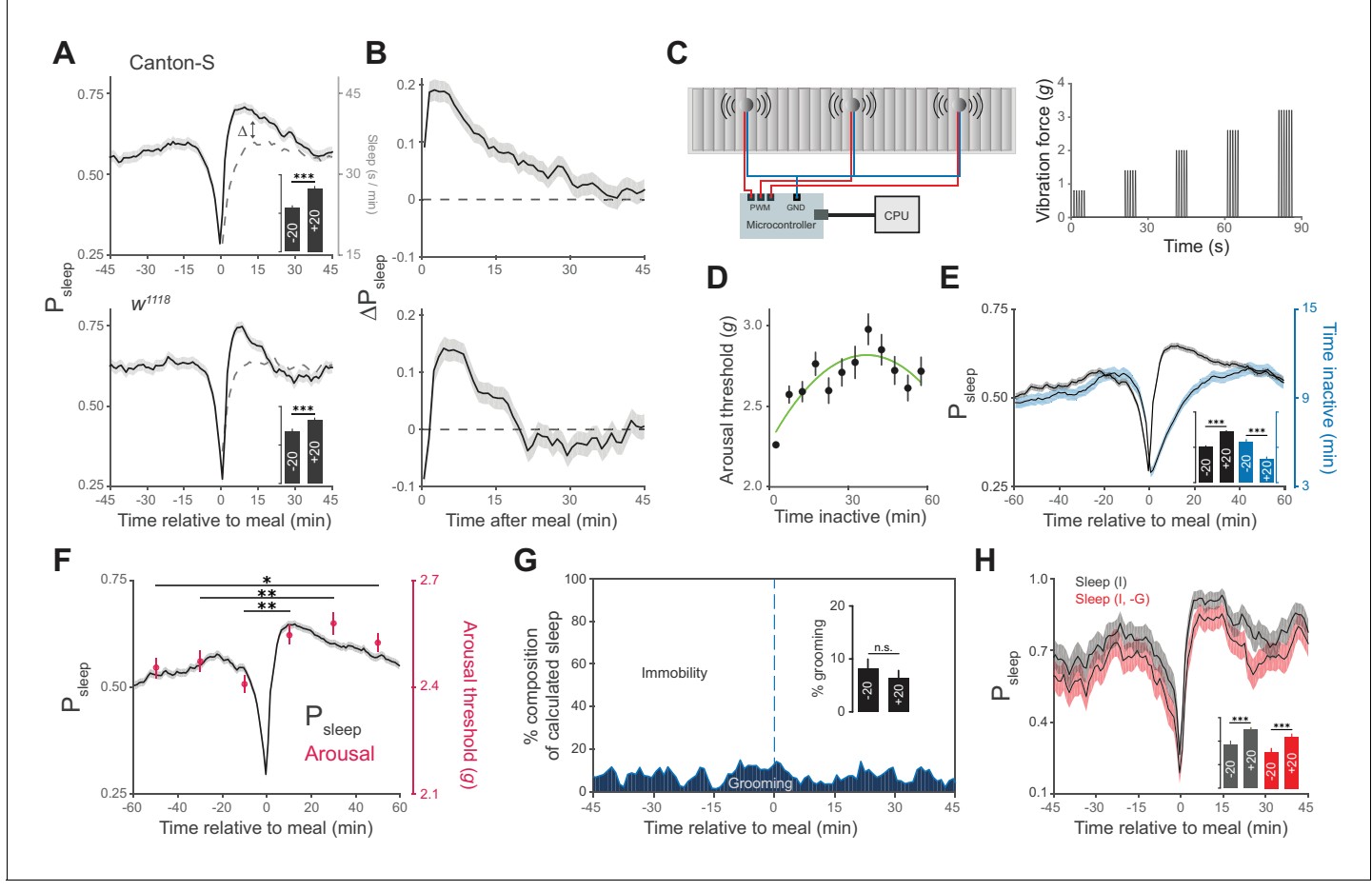

**Figure 2.** Animals exhibit increased sleep and arousal threshold after eating. (**A**) Probability of sleep ($P_{sleep}$) preceding or following each meal (t = 0) in Canton-S (top) or $w^{1118}$ (bottom) males. Data are shown as averages of 1 min bins. $\Delta P_{sleep}$ is defined as the difference between postprandial $P_{sleep}$ and the corresponding time-matched pre-meal $P_{sleep}$ (i.e. $t^1 - t^{-1}$, $t^2 - t^{-2}$, ...). For clarity, a mirror image of pre-meal $P_{sleep}$ is replotted in the postprandial period (dashed line). Inset graphs show average $P_{sleep}$ (± s.e.m.) for the 20 min before (−20) and after (+20) meals, with the axis scaled to the parent graph. n = 757 meals from 50 flies, Canton-S; 661 meals from 30 flies, $w^{1118}$; ***p<0.001, Wilcoxon matched-pairs sign rank test. (**B**) $\Delta P_{sleep}$ calculated from data in A. (**C**) Diagram of stimulus delivery system showing shaft-less vibration motors attached to the back of ARC chamber. Increasing vibrations are delivered to the chamber via a microcontroller using pulse width modulation. (**D**) Arousal threshold shows an initial increase with the time an animal is inactive. 180 flies, 11,479 arousal events, Canton-S; 5 min bins, circles represent mean ± s.e.m., a secondorder polynomial trendline is shown. (**E**) Superimposition of time inactive (blue) over $P_{sleep}$ (black) relative to meals during periodic vibrational stimuli. The inset graph shows $P_{sleep}$ and time inactive in the 20 min before and after each meal. n = 2245 meals from 180 flies, Canton-S; ***p<0.001, Wilcoxon matched-pairs sign rank test. (**F**) Stimulus response from 0–20, 20–40, and 40–60 min pre- and post-meal (red) superimposed onto $P_{sleep}$ (black). Arousal events are filtered to 5 min intervals for prior time inactive to control for sleep depth (0–5 mins shown, minimum 1 s inactivity). Circles represent mean ± s.e.m.; n = 2245 meals from 180 flies, Canton-S; *p<0.05, **p<0.01, Mann Whitney test. (**G**) Percent of calculated sleep that is actual immobility versus grooming. The inset graph shows the percent grooming in the 20 min before and after each meal. n = 55 meals from seven flies, Canton-S; p=0.69, Wilcoxon matched-pairs sign rank test. (**H**) Comparison of $P_{sleep}$ before and after meals calculated using immobility criteria "I" versus immobility criteria paired with grooming criteria "I, -G". Inset shows average $P_{sleep}$ (± s.e.m.) in the 20 min before and after each meal. ***p<0.001, Wilcoxon matched-pairs sign rank test.

The following figure supplements are available for figure 2:

**Figure supplement 1.** Postprandial sleep is sex independent.

**Figure supplement 2.** Arousal threshold surrounding meals with increasing sleep duration state.

**Figure supplement 3.** Grooming event influence on immobility based sleep.

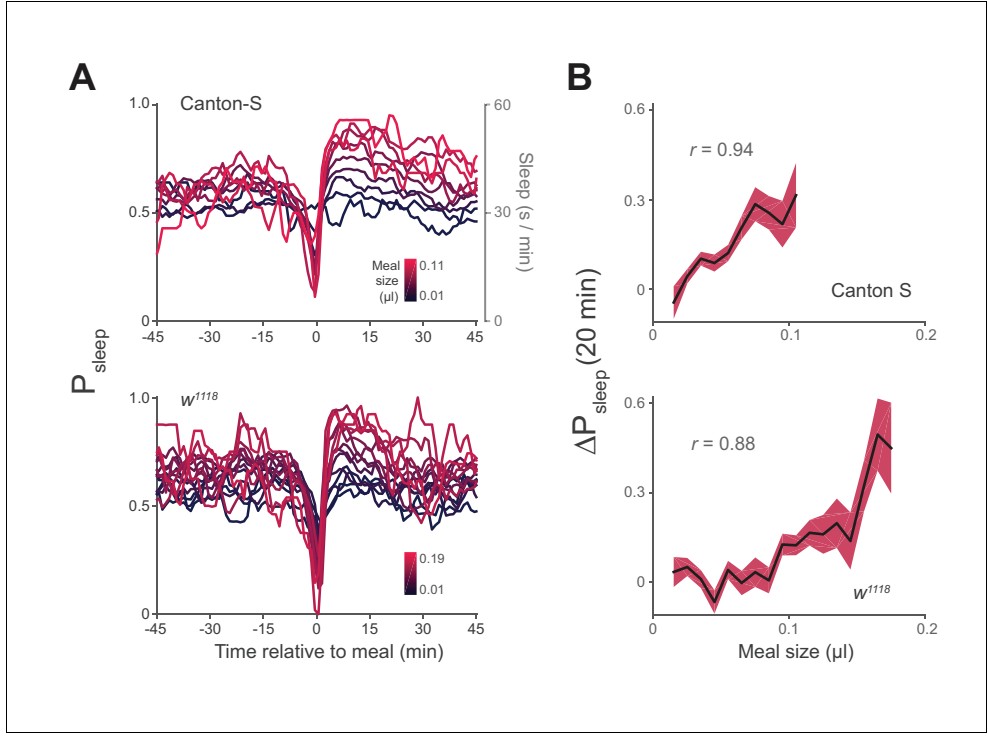

**Figure 3.** Postprandial sleep correlates with food intake quantity. (**A**) Sleep probability of Canton-S and $w^{1118}$ surrounding meals of varying size (lines represent mean; color grading corresponds to meal size). Only groupings with n > 7 are shown for visualization. (**B**) 20 min $\Delta P_{sleep}$ as a function of meal size for each grouping (Pearson correlation: p<0.001 for Canton-S and $w^{1118}$.) Shaded lines represent mean ± s.e.m.

The following figure supplements are available for figure 3:

**Figure supplement 1.** Sleep probability and $\Delta P_{sleep}$ response to meal size.

**Figure supplement 2.** Unequal meal sampling frequency and motor ability effects on meal size-sleep correlation.

with a strong but transient increase in $\Delta P_{sleep}$ amplitude (***Figure 4A***, ***Figure 4—figure supplement 2A***).

To determine if individual nutrients could modulate postprandial sleep, we fed animals a concentration series of protein (tryptone), salt (NaCl), or sucrose. All diets contained a base of 0.25% protein to induce feeding while minimizing effects on baseline sleep that might arise by differences in sucrose consumption. Within each series we compared nutrient consumption within a narrow volume range (0.02–0.04 μl, representing 33–38% of recorded meals) to limit the contribution of ingested volume on sleep. We found that $\Delta P_{sleep}$ was significantly correlated to both protein and salt ingestion (***Figure 4A***, ***Figure 4—figure supplement 1A–B***). Interestingly, despite reported effects of sucrose on total sleep (***Keene et al., 2010***; ***Linford et al., 2012***), we found no correlation between ingested sucrose and $\Delta P_{sleep}$ (***Figure 4A***, ***Figure 4—figure supplement 1A–B***). For all analyses, only the meal parameter of interest varied, while all others were kept within a restricted range (***Figure 4B***). To compare the kinetics of $\Delta P_{sleep}$ for each effective driver, we quantified maximum $\Delta P_{sleep}$ amplitude and the total $\Delta P_{sleep}$ during its decay. For any given increase in total $\Delta P_{sleep}$ during the decay, meal volume induced a greater rise in amplitude than protein or salt (***Figure 4—figure supplement 2B***). We hypothesize that multiple mechanisms operating on different time scales might underlie postprandial sleep regulation by the various drivers.

We next sought to identify a neuronal mechanism by which feeding drives postprandial sleep. A previous study showed that Lk was involved in meal size regulation, suggesting that this system acts rapidly during feeding to signal a behavioral shift (***Al-Anzi et al., 2010***). To test if Lk signaling

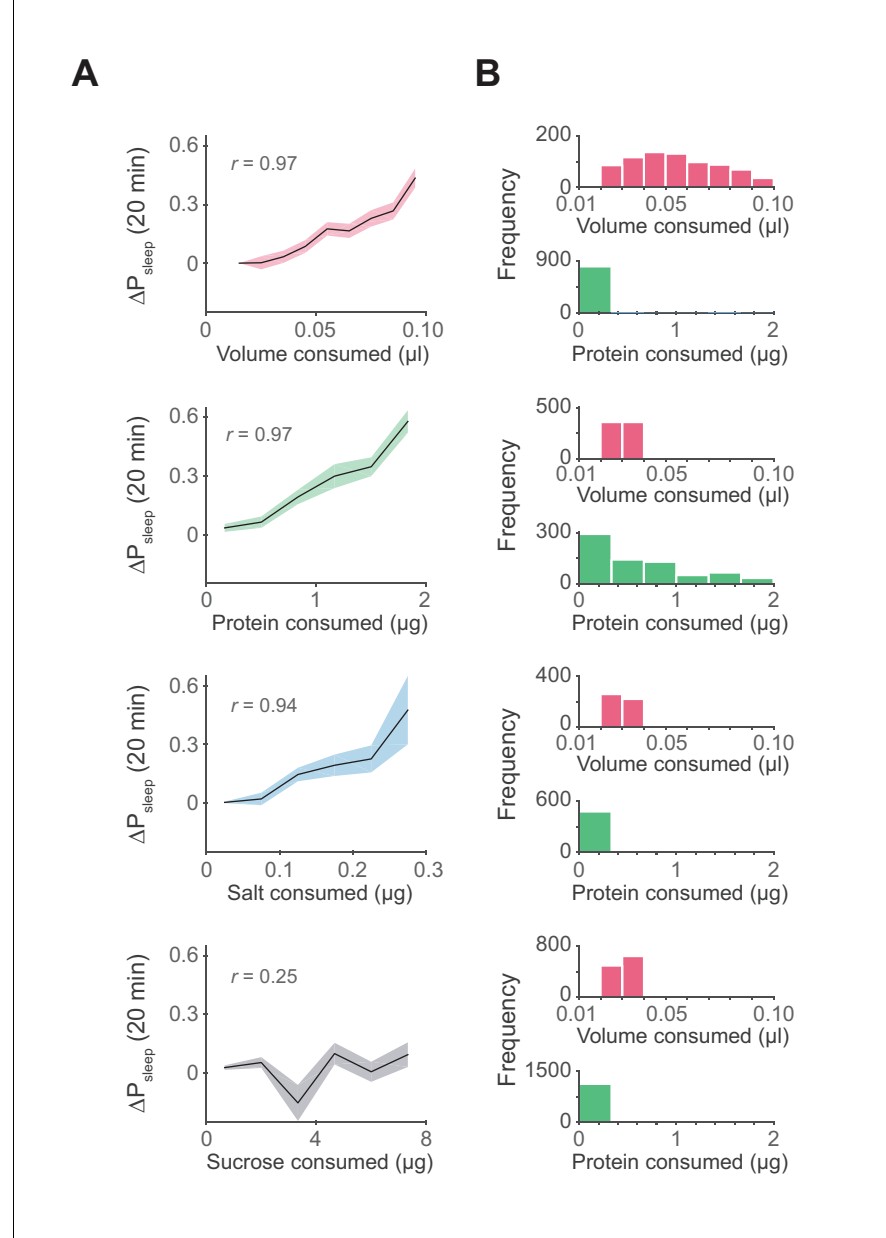

**Figure 4.** Influence of meal components on postprandial sleep. (A) Average 20 min $\Delta P_{sleep}$ as a function of each meal component (Canton-S, Pearson correlation: p<0.001, volume; p<0.001, protein; p<0.005, salt; p=0.065, sucrose). Shaded lines represent mean ± s.e.m. (B) Histograms representing the distribution of volume and protein consumed in each meal for data from A.

The following figure supplements are available for figure 4:

**Figure supplement 1.** Meal component correlates to postprandial sleep.

**Figure supplement 2.** Time-course analysis of sleep in response to meal components reveals differential kinetics.

regulates postprandial sleep, a subset of Lkr neurons labeled by the line, *R65C07* (hereafter referred to as *Lkr^GAL4*), were silenced by overexpressing *Kir^{2.1}*, an inward rectifying K+ channel (*Johns et al., 1999*). This manipulation eliminated any rise in postprandial sleep and instead caused slight arousal (*Figure 5A*). There was no significant change in postprandial activity ($\Delta_{motion}$ for 20 min after meal:

$Kir^{2.1}$ / $w^{CS}$, $-0.006 \pm 0.003$; $Lkr^{GAL4}$ / $w^{CS}$, $-0.001 \pm 0.002$; $Kir^{2.1}$ / $Lkr^{GAL4}$, $0.004 \pm 0.004$; p=0.28, Kruskal-Wallis test followed by Dunn's multiple comparisons). The expression pattern of $Lkr^{GAL4}$ recapitulates that of Lkr neurons which were found to regulate meal size (**Al-Anzi et al., 2010**),

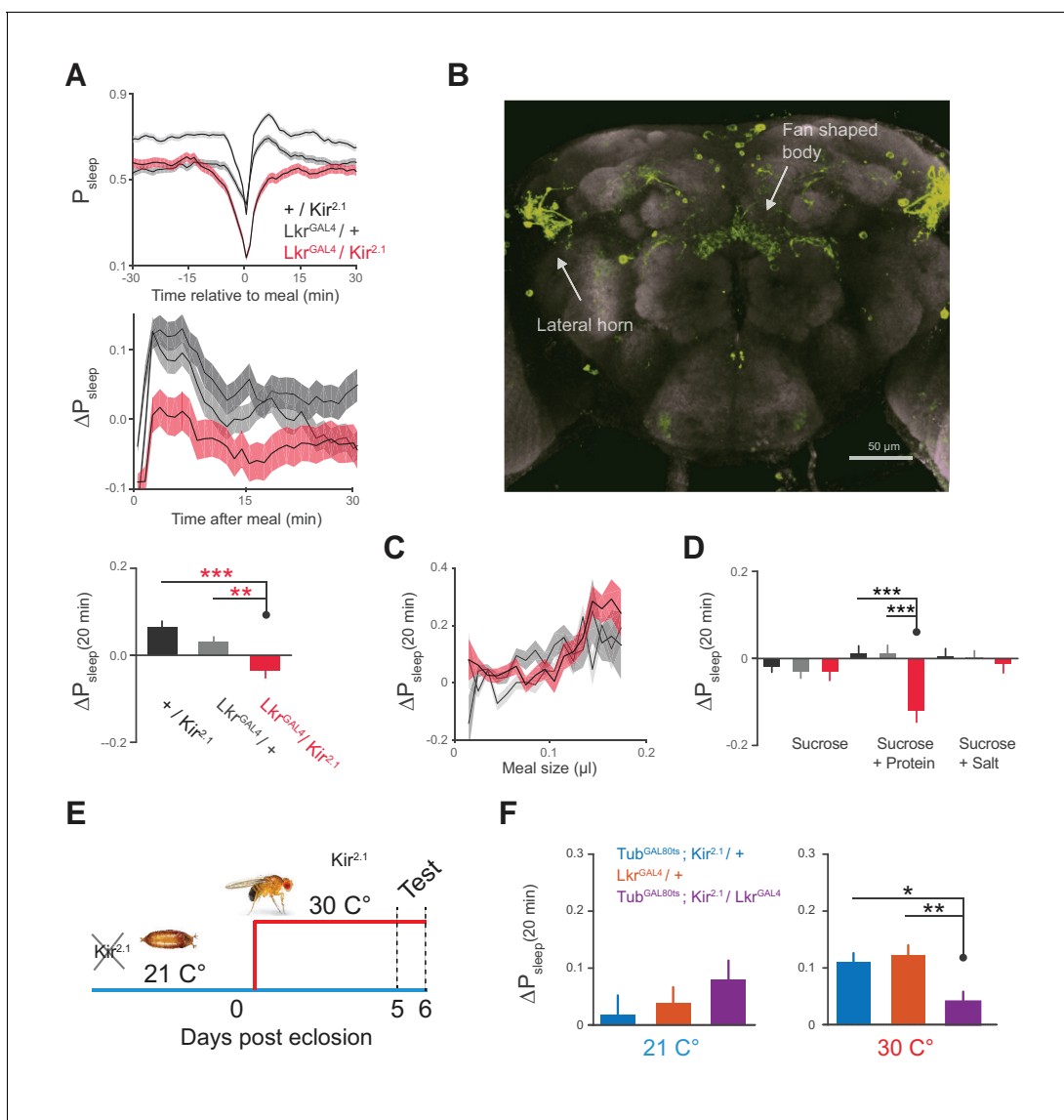

**Figure 5.** Leucokinin receptor neurons regulate protein-induced postprandial sleep. (**A**) Overexpression of $Kir^{2.1}$ channel in cells labeled by $Lkr^{GAL4}$ ($R65C07$) results in a defective postprandial sleep response (20 flies per genotype, n = 697 meals, $Kir^{2.1}$ / $w^{CS}$; 762, $Lkr^{GAL4}$ / $w^{CS}$; 450, $Kir^{2.1}$ / $Lkr^{GAL4}$; *p<0.05; ***p<0.001, Kruskal-Wallis test followed by Dunn's multiple comparisons). (**B**) $Lkr^{GAL4}$ drives mCD8-GFP expression in neurons innervating the dorsal fan-shaped body, stemming from cell bodies in the lateral horn (scale bar = 50 µm). (**C**) Average 20 min $\Delta P_{sleep}$ for $Kir^{2.1}$-silenced $Lkr^{GAL4}$ animals and controls given low nutrient food (1% sucrose, 0.25% tryptone) for observing volumetric effects (n = 1589–1736 meals per genotype). (**D**) Average 20 min $\Delta P_{sleep}$ for $Kir^{2.1}$-silenced $Lkr^{GAL4}$ animals and controls given salt (2.5% sucrose + 1% salt) or protein (2.5% sucrose + 1.7% tryptone) diet to test meal component influences on postprandial sleep. The silenced line shows a negative response to protein supplemented diet (***p<0.001, Kruskal-Wallis test followed by Dunn's multiple comparisons). n = 257–444 meals per genotype. (**E**) Expression of $Kir^{2.1}$ can be restricted by using temperature-sensitive GAL80[ts] to suppress GAL4 activity at 21°C throughout development. (**F**) Silencing of $Lkr^{GAL4}$-labeled cells in adulthood (30°C) is sufficient to reduce postprandial sleep (24 flies per genotype, n = 493 meals, $Tub^{GAL80ts}$; $Kir^{2.1}$/ $w^{CS}$; 499, $Lkr^{GAL4}$/ $w^{CS}$; 528, $Tub^{GAL80ts}$; $Kir^{2.1}$/ $Lkr^{GAL4}$; *p<0.05; **p<0.01, Kruskal-Wallis test followed by Dunn's multiple comparisons).

The following figure supplement is available for figure 5:

**Figure supplement 1.** General behavior in Lkr neuronal-silenced animals.

revealing several cell populations stemming from the lateral horn which innervate the dorsal fan-shaped body of the central complex—a region of the brain which functions in sleep homeostasis, visuomotor function, and hunger-related behavior (*Figure 5B*) (*Donlea et al., 2011, 2014*; *Seelig and Jayaraman, 2013*; *Krashes et al., 2009*). This circuitry suggests that the Lk system may act directly on the sleep and motor controllers of the fly brain to regulate postprandial sleep.

We hypothesized that Lkr circuitry could receive convergent signals from volume, protein, and salt to drive postprandial sleep. Alternatively, it could specifically relay the signals of a single driver to some point of convergence. We first tested whether the subset of Lkr neurons labeled by $Lkr^{GAL4}$-mediated the sleep response to meal volume by feeding a low nutrient diet to $Lkr^{GAL4}$-silenced animals. Silenced animals and their controls showed similar $\Delta P_{sleep}$ in response to ingested volume, indicating that the $Lkr^{GAL4}$ neurons are not responsible for the volumetric modulation of postprandial sleep (*Figure 5C*). To test the role of Lkr neurons in the postprandial sleep response mediated by nutrients, we supplemented a sucrose diet with either salt or tryptone. $Lkr^{GAL4}$-silenced animals and their controls showed statistically indistinguishable $\Delta P_{sleep}$ responses to salt manipulation (*Figure 5D*). In contrast, while protein ingestion increased postprandial sleep in controls, $Lkr^{GAL4}$-silenced animals showed a strong waking response, or a negative shift in postprandial sleep (*Figure 5D*). Interestingly, this waking response exceeded that of control animals on sucrose alone, indicating that protein can also drive wakefulness in the absence of $Lkr^{GAL4}$ neuronal activity. We did not observe changes in meal size, total sleep, or total consumption, indicating that these behaviors are independent of Lkr influence on postprandial sleep (*Figure 5—figure supplement 1A–C*). To test whether this phenotype was caused by changes in development, we conditionally silenced the cells by co-expressing a temperature sensitive GAL4 suppressant protein, GAL80$^{ts}$, throughout development (*Figure 5E*). Inactivation of GAL80$^{ts}$ to silence $Lkr^{GAL4}$-labeled cells in adulthood was sufficient to induce a postprandial sleep defect (*Figure 5F*).

Subsets of Lk neurons have been found in the lateral horn, with the diffuse puncta in close proximity to the cell bodies of the Lkr neurons, suggesting a neuromodulatory connection between these cells (*Al-Anzi et al., 2010*; *de Haro et al., 2010*; *Cavey et al., 2016*). Recent findings also show that incubating Lkr neurons with Lk inhibits their response to the cholinergic agonist carbachol, suggesting that Lk inhibits Lkr neuronal activity (*Cavey et al., 2016*). We first sought to confirm that cells labeled by the $Lk^{GAL4}$ driver express Lk. Indeed, immunostaining of Lk overlapped with $Lk^{GAL4}$-labeled cells in the lateral horn, as well as in two cells in the suboesophageal ganglion (SOG) (*Figure 6A*). Using thermogenetic activation of these cells with the transient receptor potential channel Trp$^A$, we found that postprandial sleep was significantly reduced upon activation (*Figure 6B*), furthering the notion that these cells inhibit Lkr neurons. To see if Lk was necessary to drive this inhibitory effect, we used two independent RNAi lines to knockdown Lk in cells labeled by $Lk^{GAL4}$. We found that this knockdown increased postprandial sleep, consistent with the idea that expression of Lk in $Lk^{GAL4}$-labeled cells is necessary for the inhibitory effect on Lkr neuronal activity (*Figure 6C*).

The broad arborizations of the Lkr neurons suggest that these cells might receive inputs from a number of spatially or type distinguished cells in the brain. *R20G03* (*20G03*), a line expressing GAL4 driven by an enhancer fragment 316 base pairs upstream of Lk, labels cell bodies in the lateral horn and SOG, as well as a subset of mushroom body output neurons (MBONs) (*Figure 7A*), all of which coincide with Lkr neurons labeled by the $Lkr^{GAL4}$driver (*Figure 7B*). The cholinergic MBONs of *20G03* have been shown to play a role in short-term appetitive memory following food intake (*Aso et al., 2014*). These features, paired with their neuroanatomical distribution, suggested that these cells might play a role in postprandial sleep. To test this, we again employed electrical silencing with $Kir^{2.1}$ and found that constitutive and adult-restricted silencing of these neurons increased postprandial sleep, suggesting that these cells might directly inhibit the activity of proximal Lkr neurons (*Figure 7C–D*). Interestingly, immunostaining of Lk did not co-localize with any of the *20G03* cells, nor did downregulation of Lk in these cells affect postprandial sleep (*Figure 7—figure supplement 1A–B*), suggesting that their inhibitory action stems from an alternative neuropeptide or neurotransmitter. By analyzing behavior throughout a 24 hr period, we found that the increase in postprandial sleep driven by *20G03* silencing was most pronounced at ZT 12 (*Figure 7E–F*). Closer examination within this period showed that, rather than raising sleep evenly within ZT 10–14, the sleep after but not before each meal was increased (*Figure 7G*), implying that these cells activate in response to food intake.

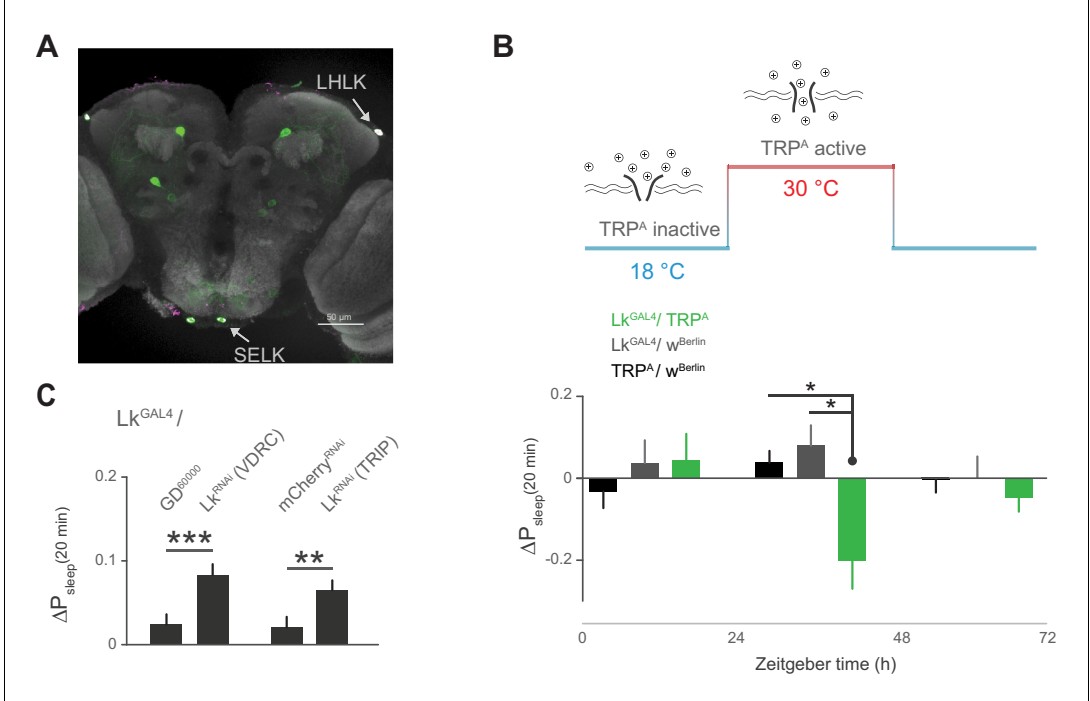

**Figure 6.** Leucokinin neurons inhibit postprandial sleep. (**A**) Confocal reconstruction of immunostaining for anti-Lk (magenta) in the brain of $Lk^{GAL4}$>mCD8::GFP reveals Lk co-localization with GFP-expressing neurons (green) in the lateral horn (LHLK) and suboesophageal ganglion (SELK). The neuropil marker nc82 (gray) is used as background (scale bar = 50 μm). (**B**) Stimulation of Lk neurons at 30°C by expressing Transient receptor potential channel, $Trp^A$, causes a reduction in postprandial sleep in comparison to the unstimulated state at 18°C (10 flies per genotype, 30°C, n = 83 meals, $Trp^A$/ $w^{Berlin}$; 63, $Lk^{GAL4}$ / $w^{Berlin}$; 25, $Lk^{GAL4}$ / $Trp^A$; *p<0.05, Kruskal-Wallis test followed by Dunn's multiple comparisons). (**C**) Downregulation of Lk in $Lk^{GAL4}$-labeled cells, using two independent RNAi lines, increases postprandial sleep. (60 flies per genotype, n = 893 meals, $Lk^{GAL4}$ / $GD^{60000}$; 841, $Lk^{GAL4}$ / $Lk^{RNAi}$ (VDRC); 953, $Lk^{GAL4}$ /$mCherry^{RNAi}$; 1060, $Lk^{GAL4}$ / $Lk^{RNAi}$ (TRIP); **p<0.01, ***p<0.001 Mann-Whitney test).

## Discussion

The ARC is the first platform capable of simultaneously measuring sleep and resolving discrete feeding events in individual flies. Using this tool, we found that flies exhibit a rise in sleep following a meal. Interestingly, consumption of a meal did not necessarily affect the depth of sleep when controlling for time inactive. There are many instances in which genes or neural processes have been found to influence particular features of sleep while leaving others unaffected (*Harbison et al., 2009*; *Shi et al., 2014*; *Shang et al., 2011*). Similarly, our work suggests that sleep probability and sleep depth can also be dissociated. By further examination of the shift in sleep probability, we found that it increased as a function of meal size. This may be paralleled in humans where cranial EEG power increases with meal size and during certain stages of sleep (*Reyner et al., 2012*).

In examining individual meal components, we identified volume, protein, and salt as effectors of postprandial sleep. While human studies have been unable to resolve a link between meal volume and measures of wakefulness (*Landström et al., 2001*), artificial distension of the gut has been shown to increase sleep in rats (*Lorenz et al., 1998*). Although there are no studies on the effect of protein on postprandial sleep, protein consumption modulates sleep in *Drosophila* and positively correlates with napping in humans (*Catterson et al., 2010*; *Grandner et al., 2010*). Similarly, although there are no studies on the influence of salt on postprandial sleep, salt intake increases mammalian oxytocin signaling, which has been implicated in sleep-wake behavior (*Krause et al., 2011*; *Lancel et al., 2003*). Sucrose has been shown to play a critical role in long-term sleep homeostasis and architecture in flies (*Keene et al., 2010*; *Linford et al., 2012*), although our findings indicate that it does not modulate postprandial sleep. Although we demonstrate the effectiveness of these dietary components in isolation, we note that meals typically incorporate multiple elements. Thus, further work is necessary to identify the mechanisms by which each constituent is sensed and

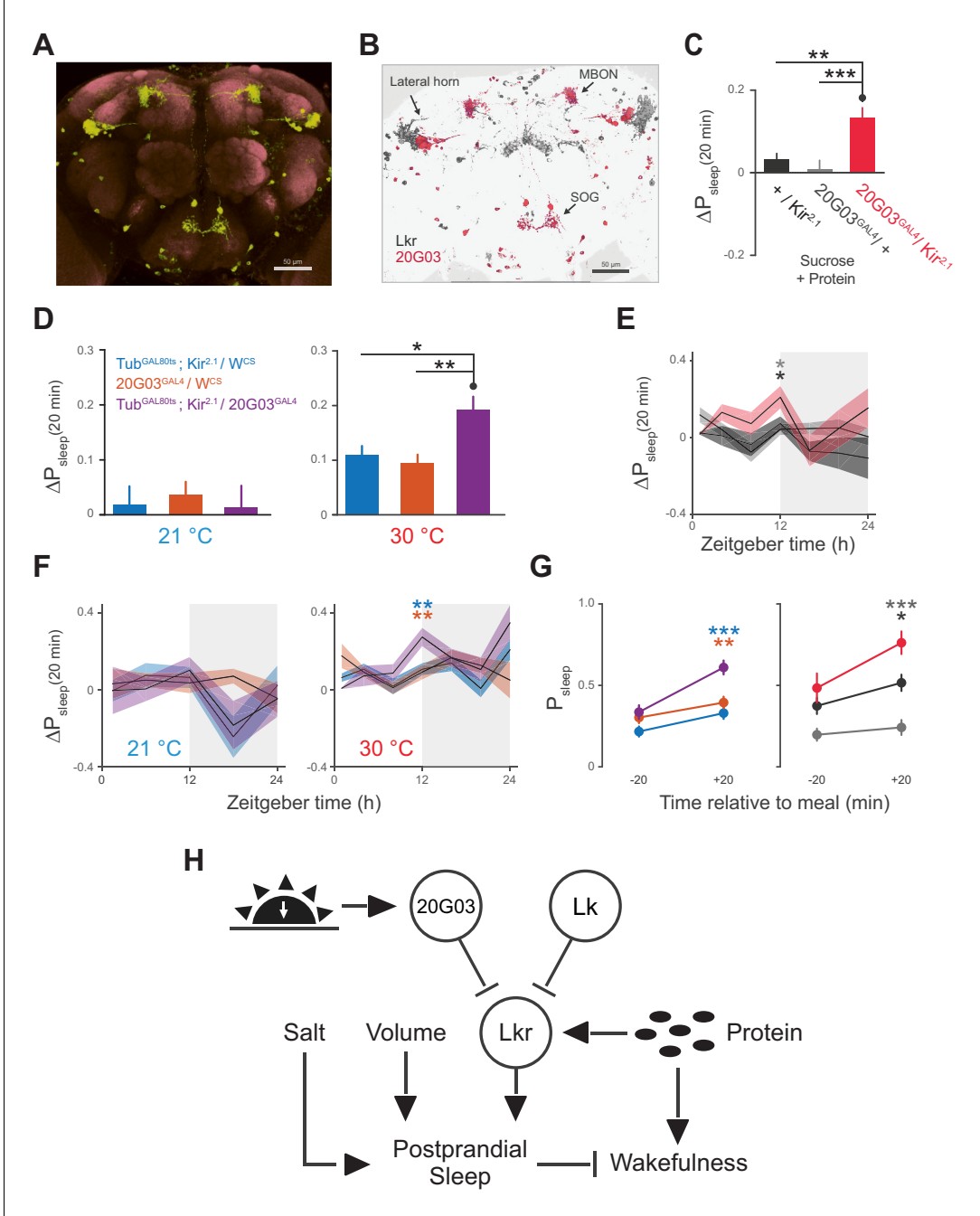

**Figure 7.** *20G03* neurons inhibit postprandial sleep in a circadian manner. **(A)** *20G03*[GAL4] drives mCD8-GFP expression in neurons with cell bodies positioned proximal to cell arborizations of those labeled by *Lkr*[GAL4]. **(B)** Overlaid images of *20G03*[GAL4] and *Lkr*[GAL4] expression show close proximity between cell types in the lateral horn and SOG. **(C)** Overexpression of $Kir^{2.1}$ channel in *20G03*[GAL4]-labeled cells results in an enhanced postprandial sleep response (30 flies per genotype, n = 805 meals, $Kir^{2.1}$ / $w^{1118}$; 488, *20G03*[GAL4] / $w^{1118}$; 285, $Kir^{2.1}$ / *20G03*[GAL4]; **p<0.01; ***p<0.001, Kruskal-Wallis test followed by Dunn's multiple comparisons). **(D)** Conditional silencing of *20G03*[GAL4]-labeled cells in adulthood is sufficient to increase postprandial sleep (24 flies per genotype, n = 493 meals, *Tub*[GAL80ts]; $Kir^{2.1}$/ $w^{CS}$; 575, *20G03*[GAL4] / $w^{CS}$; 528, *Tub*[GAL80ts]; $Kir^{2.1}$/*20G03*[GAL41]; *p<0.05; **p<0.01, Kruskal-Wallis test followed by Dunn's multiple comparisons). **(E)** Time-course of postprandial sleep reveals that effect of *20G03*[GAL4] silencing is most prominent from ZT 10–14 (time-course partitioned into 4 hr bins, shaded region indicates dark period; *p<0.05, Kruskal-Wallis test followed by Dunn's multiple comparisons). **(F)** Conditional silencing effects are also stronger at dusk ZT 10–14 (**p<0.01, Kruskal-Wallis test followed by Dunn's multiple comparisons). **(G)** Comparison of $P_{sleep}$ in the 20 min before and after each meal for both *20G03*[GAL4] manipulations (color codes from C and D; *p<0.05; **p<0.01, ***p<0.001, Kruskal-Wallis test followed by Dunn's multiple comparisons). **(H)** Proposed model for regulation of postprandial sleep by dietary components and neuronal circuitry. Meal volume, ingested salt, and protein drive postprandial sleep. Sleep induced by ingested protein acts

*Figure 7 continued on next page*

*Figure 7 continued*

through Lkr neurons. Protein also induces a waking response independent of Lkr neuronal activity. Leucokininergic (Lk) or non-leucokininergic (20G03) cell populations can independently inhibit postprandial sleep, possibly through modulation of Lkr neuronal activity.

The following figure supplement is available for figure 7:

**Figure supplement 1.** Lk immunostaining and knockdown in *20G03* neurons.

the potential interactions between them. Lipids are also present in many fly food sources and future examination of their influence on postprandial sleep might be facilitated by adding fat solubilizers to the liquid food used in the ARC.

We demonstrate that the Lk system plays a role in postprandial sleep (*Figure 7H*). A subset of Lkr neurons were necessary to initiate postprandial sleep in the presence of protein. While we expected that animals defective in protein sensing would experience postprandial sleep similar to that of animals fed only sucrose, we found instead that they had a waking response. This suggests that ingested protein promotes both sleep and wakefulness, and that the wakefulness is normally counterbalanced by Lkr neuronal activity. While this study does not specifically identify the waking output for protein, it was previously shown that Translin acts through Lk-expressing cells to drive wakefulness in response to sucrose starvation (*Murakami et al., 2016*). While this does not explain rapid wake promotion in response to protein ingestion, it does demonstrate that Lk-expressing cells are capable of promoting wakefulness.

Another recent finding identified Lkr neurons as circadian output neurons, which modulate activity rhythmically. The same study found that incubation of Lkr cells with Lk shunts their calcium response to the cholinergic agonist carbachol (*Cavey et al., 2016*), suggesting an inhibitory connection between Lk and downstream Lkr neurons. By stimulating these same Lk neurons, we observed a reduction of postprandial sleep matching the effect of Lkr silencing, providing additional evidence that the lateral horn harbors an inhibitory node between Lk and Lkr neurons. We also found that knockdown of Lk in these cells increased postprandial sleep, supporting this notion. However, it has been suggested that Lk may have additional receptors (*Terhzaz et al., 1999*), and there is still a lack of definitive evidence for the inhibitory mechanism between Lk and Lkr neuronal activity in the lateral horn. Additional genetic tools, including lines with restricted expression patterns, will be needed to fully uncover the genetic interactions and neuronal connectivity of this system.

We also found a subset of cells labeled by *20G03* that modulate postprandial sleep during the period near 'dusk' or 'lights-off'. Interestingly, the subset harbors acetylcholine MBONs—cells necessary for the short-term associative memory between sucrose consumption after starvation and an odor (*Aso et al., 2014*). This suggests that these cells activate following a feeding event, in agreement with our results. This is further supported by the anatomical distribution of cells in the SOG, a region of the fly brain that regulates feeding behavior. The distribution also shows overlap with Lkr neurons and regions of the neuronal clock network, suggesting a diversity of regulatory inputs and outputs. Further studies might employ GFP reconstitution across synaptic partners (GRASP) (*Gordon and Scott, 2009*) or neuronal co-labeling to demonstrate these connections. The apparent diversity of neuronal modulators of postprandial sleep opens the door to future work on determining the various genes and circuits involved.

The tachykinin family to which Lk belongs has been implicated in the regulation of satiety and sleep, however any physiological connections between these behavioral outputs has not been explored (*Massi et al., 1988*; *Zielinski et al., 2015*). The orexin system is regulated by a diversity of signaling molecules (*Scammell and Winrow, 2011*) and has already shown promise as a target for sleep therapeutics. Such genetic diversity and therapeutic potential may also be true of systems governing postprandial sleep. Accordingly, the ARC provides a platform for future studies aimed at uncovering additional genes, circuitry, and physiological roles of postprandial sleep. Furthermore, the ability to measure both sleep and feeding in individual animals may shed new light on the fundamental relationship between these behaviors in other paradigms.

## Materials and methods

### *Drosophila* lines and culture

Flies were developed in 6 oz. bottles on a standard cornmeal-sucrose-yeast medium consisting of (w/v) 5.8% cornmeal, 3.1% active dry yeast, 0.7% agar, 1.2% sucrose, and (v/v) 1% propionic acid and 0.22% Tegosept (w/v, pre-dissolved in ethanol). Typically, males were collected 0–2 days after eclosion under gentle $CO_2$ anesthesia and maintained on standard diet at ~20 flies per vial (Polystyrene, 25 × 95 mm). All flies were maintained in a humidity and temperature controlled incubator at 25°C and 65% relative humidity under a 12/12 hr light/dark cycle. All experiments were carried out with 5–9 day old males. *R20G03* (*20G03*, RRID:BDSC_48907) and *Lkr*$^{GAL4}$ (*R65C07*, RRID:BDSC_39344) neuronal driver expression patterns were identified by and obtained from Janelia Farm (*Jenett et al., 2012*) via the Bloomington stock center. *UAS-Kir*$^{2.1}$(*Baines et al., 2001*) and *Tub*$^{GAL80ts}$ (*McGuire et al., 2004*) were previously described. *Lk*$^{RNAi}$ (TRIP, RRID:BDSC_25798) was obtained from the Bloomington Stock Center. *Lk*$^{RNAi}$ (VDRC, 14091) was obtained from the Vienna *Drosophila* Resource Center. *UAS-TRP*$^{A}$ was provided by Ulrike Heberlein. *Lk*$^{GAL4}$ and the anti-Lk antibody were provided by Bader Al-Anzi.

### Diets

Liquid diets were prepared by boiling ingredients in 100 ml dd$H_2$O followed by filtration (0.2 µm cellulose acetate syringe filter, VWR, Radnor, PA). Listed percentages of Bacto Tryptone, Bacto yeast extract, sucrose, or NaCl were calculated as weight/volume. All reagents were obtained from Fisher Scientific (Waltham, MA) or VWR. Feeding behavior was recorded on a standard liquid diet of 2.5% sucrose + 2.5% yeast extract unless otherwise noted.

### ARC behavioral chamber

Animals were housed in a plastic chamber (acrylonitrile butadiene styrene) containing small half-cylindrical cells (7 mm wide, 4.5 mm depth, 1.15 mm spacing) fit with a 2 mm thick clear acrylic panel for visualization. Strips of infrared led lights were placed ~16 cm behind each chamber on a heat sink to provide backlighting for feeding and motion tracking. Each animal was given access to a capillary filled with liquid food. Above the chambers was a 2 mm gap with a small overhang allowed for a sliding gate (5 mm × 2 mm) to seal individual fly cells. The gate contained evenly spaced 2.5 mm vertically oriented holes above the center of each fly cell. This allowed for the insertion of tightly fitting 200 µl pipette tip sleeves that were cut to hold the food-containing capillaries (5 µl glass capillaries, VWR). Sleeves were made by cutting the narrow end of 200 µl pipette tips to fit capillaries tightly. The dye used for liquid meniscus tracking was composed of 75% mineral oil, 25% dodecane, and 1% Copper (II) 1,4,8,11,15,18,22,25-octabutoxy-29*H*,31*H*-phthalocyanine (Sigma-Aldrich, St. Louis, MO). Dye solution was vortexed for 1 min and centrifuged briefly. Supernatant dye was collected and re-dispensed into 1.5 ml tubes and vortexed for 1 min. Dye solution was loaded into each capillary to make a 1 mm plug before loading liquid diet. Food was loaded until dye bands were approximately 0.5 mm below the capillary reference mark. To improve visualization of the dye and reference mark, thin white construction paper was gently attached to the back of capillaries using double-sided tape.

### Feeding behavioral capture and analysis

For ARC serial image capture, we used a Lifecam 1080p HD webcam (Microsoft, Redmond, WA) with the infrared filter replaced by an infrared pass filter. The camera was mounted with a custom bracket at equal height to the base of the capillaries. PhenoCapture software was used to capture one image per min using the time-lapse image capture tool (www.phenocapture.net). Each capillary had an exterior reference mark and was pre-loaded with an oil-based dye band, marking the meniscus of the liquid food. ImageJ was used for image processing (*Schneider et al., 2012*). Images were cropped to include all capillaries and dye bands. The images were background subtracted using the built-in subtract background function with a rolling ball radius of five and disabled smoothing. The images were then thresholded manually until the dye bands and reference marks were faithfully displayed without background noise. The images were split to contain image stacks of individual capillaries and tracked using a custom ImageJ plugin. The relative distance between the food

meniscus and the reference mark was calculated at each frame using the custom plugin. We used a meal selection algorithm that estimates noise by an iterative filtration of distal values 4.0 s.d. above the $\Delta_{pixel}$ mean (see *Supplementary file 1*). Meals were then selected as values which were 3.5 s.d. above the estimated noise, based on the one frame/min sampling rate. In a setup tracking 30 capillaries, this method provides 10 nL resolution although greater resolution can be achieved by increasing camera resolution and sampling rate. Though infrequent, temporally adjacent feeding bouts were combined and time stamped with the latter meal time. Thus, feeding bouts followed by at least 1 min of non-feeding were considered meals. Each feeding event was corrected for effects of evaporation and luminance fluctuation on observed meal size by subtracting the average noise value from all non-feeding frames. Meal output contains meal associated time and corrected volume.

## Positional tracking

Behavioral tracking was carried out with a custom, freeware, cross-platform analysis framework (JavaGrinders Library, available for free download at http://iEthology.com/). This library utilizes and extends machine vision functions provided by the OpenCV project for high-resolution analysis of spatial and behavioral data. With an integrated interface for standard USB video device class (UVC) cameras, the software implements multi-stage object detection and analysis for 8-bit grayscale/32-bit color frame sequences in real-time. Following the subtraction of a reference frame, the tracker locates the frame coordinate with the highest remaining pixel value. Provided this value exceeds the object's threshold, the detection algorithm proceeds outwards until it identifies a starting point along the object's thresholded border. By following this edge until it returns to the starting point, the detection algorithm characterizes the object's outline as a polygonal shape. The object's center location is estimated by the polygon's centroid, directional attributes (e.g., long axis) which are obtained via a Singular Value Decomposition of the polygon's outline. The projected size is represented by the shape's area. The size of a fly tracked in the ARC measured approximately 15–20 pixels in length. The present study used a Microsoft Lifecam Studio camera with the infrared filter replaced by an infrared pass filter. Although much higher rates are possible within this system, a reduced sampling rate of 1 Hz was chosen as it proved sufficient for identifying the extended periods of inactivity indicative of sleep. The maximum frame rate capability depends on a number of factors, including the video's resolution, the object's size, the performances of processor and graphics card, and the specifications of the hardware drivers used to interface with the camera. Distance traveled was obtained from Euclidean measures between object centers in consecutive frames. Motion values less than 50% of the fly body length (FBL) were discarded. Dropped positions were reset to the last known location for motion and sleep calculation purposes. This system has been previously tested for *Drosophila* tracking sufficiency and demonstrated enhanced motion/sleep characterization compared to alternative methods (*Donelson et al., 2012*). An animal was considered sleeping during any period of immobility exceeding 5 min, based on previous studies showing that this period of inactivity is highly correlated with hallmarks of sleep (*Shaw et al., 2000*; *Huber et al., 2004*). Speed of each motion event was calculated as the distance traveled during that event divided by its uninterrupted duration. This is the first time, to our knowledge, that sleep has been measured in vertically oriented chambers, and on a liquid diet. However, we found sleep patterning in the ARC to be in line with the more commonly used Drosophila Activity Monitor (TriKinetics, Waltham, MA). Notably, object detection provides a higher resolution for sleep bout duration (*Figure 1—figure supplement 1D*).

## ARC experimental design

Each fly well was loaded with 300 μl of 1% (w/v) Bacto agar to allow *ad libitum* access to water. Flies were loaded into chambers using a standard mouth aspirator to avoid behavioral perturbation from $CO_2$ anesthetization. After loading each fly, a sleeve and capillary were quickly inserted to prevent escape. Flies were habituated in the recording chamber for 20–24 hr with access to high nutrient food (5% sucrose + 5% yeast extract) to obviate the need for more than one food change every 24 hr. All experiments had daily food changes from Zeitgeber time (ZT) 0–0.5 hr. Infrequently, food consumption exceeded the volume administered requiring an additional food change at ZT 8–12.

## Behavioral data analysis

Motion and feeding data were analyzed using custom python-based software and custom or built-in Matlab functions (The Mathworks, Natick, Massachusetts, USA). Instructions and 3D print files for ARC setup and data analysis software are available upon request.

Probability of sleep is calculated as the population mean of sleep probability/unit time. 1-min bins were used for all $P_{sleep}$ plots. $\Delta P_{sleep}$ represents a comparison of pre-meal to post-meal sleep by calculating the difference in $P_{sleep}$ between post-meal time bins and their respective pre-meal time bins (i.e. ($t^{0-1}$ min post-meal) – ($t^{0-1}$ min pre-meal), ($t^{1-2}$ min post-meal) – ($t^{1-2}$ min pre-meal). . .). This was either calculated in 1-min bins for time-course analysis, or in 20-min time bins following the meal. The 20-min bin was selected since this length of time covered the mean duration of postprandial sleep for both Canton-S and $w^{1118}$.

## Arousal threshold analysis

To test arousal threshold, we adapted methodology previously used for quantifying sleep and sleep depth (*van Alphen et al., 2013*; *Faville et al., 2015*). Animals were exposed hourly to a series of vibrations of increasing intensity ranging from 0.8 to 3.2 *g*, in steps of 0.6 *g*. Stimuli trains were composed of 200 ms vibration with 800 ms inter-vibration interval and 15 s inter-stimuli train interval. Stimulation intensity and timing were controlled using pulse-width modulation via an Arduino UNO and shaft-less vibrating motors (Precision Microdrives, model 312–110) (*Figure 2C*). Arousal to a given stimulus was assigned when an animal (1) was inactive at the time of the stimulus, (2) satisfied a given inactivity criteria at the time of the stimulus, and (3) moved within the inter-stimuli train period (15 s) of that stimulus.

## Grooming analysis

To determine the impact of grooming behavior on calculated sleep, we simultaneously recorded video of flies during a normal ARC experiment using equipment and videocapture methods previously described (*King et al., 2016*). Grooming start and stop times were manually noted, where the observer was blind with respect to feeding times. Times were then computer annotated as a binary (grooming = 1, not grooming = 0) for all time points within the experiment. Individual housing precluded the need for annotation of courtship behaviors.

## Meal size, volume, and nutrient grading

For meal volume gradings, animals were fasted for 20 hr and then placed on a 0.25% tryptone diet. Feeding events were recorded for 24 hr. This paradigm provided a high range of feeding volumes while maintaining a small but sufficient amount of nutrients to elicit feeding. Meal volumes and associated sleep events were grouped into 0.01 μl bins according to volume consumed. Bins greater than the 97th percentile were excluded to maintain sufficient sample size for each bin. Alternative experiments for testing volumetric effects used small amounts of sucrose (1%) or both sucrose and tryptone (1% sucrose + 0.25% tryptone) and gave similar results (data not shown). For sucrose grading, the same paradigm was used with a low tryptone diet (0.25%) supplemented with varying concentrations of sucrose (0, 1, 5, 15, and 25%). Feeding events were then filtered to include volumes within the range 0.02–0.04 μl which included 37% of all recorded meals (1131/3038 meals). This range was chosen to maintain a small and narrow volumetric input, and because all dietary groups maintained meal volumes within this range. Meals were grouped into 1.333 μg bins of sucrose consumed during the meal (meal volume × nutrient concentration). Salt grading was performed similarly but with varying concentrations of salt (0, 0.25, 0.3, 0.4, 0.5, 0.6, 0.75, and 1%) and 0.05 μg bins, and included 33% of all recorded meals (478/1446 meals). Protein grading was performed by providing diets of varying concentrations of tryptone (0.25, 1, 2.5, 3, 4, and 5%) and analyzed similarly to sucrose and salt experiments using 0.333 μg bins, and included 38% of all recorded meals (714/1872 meals).

## $Lkr^{GAL4}$ dietary manipulations

For $Lkr^{GAL4}$ silencing experiments, flies were fed standard liquid diets: 2.5% sucrose + 2.5% yeast extract, respectively. To test volume response deficiency, animals were fed a low nutrient (1% sucrose + 0.25% tryptone) food which elicited a large range and frequency of meals. To test protein

response deficiency, animals were fed 2.5% sucrose supplemented with the standard diet equivalent of protein (1.67% tryptone). The standard diet contained a negligible amount of salt (with respect to inducing a postprandial sleep response). To test a salt response deficiency, animals were fed 2.5% sucrose supplemented with 1% NaCl. Average meal size between groups did not differ—hence, meal size filtering was not applied.

## Immunohistochemistry

Brains were imaged as described previously (*Murakami et al., 2016*).

## Statistics

Statistical analysis was performed using Matlab statistical toolbox or GraphPad Prism version 5.04 (GraphPad Software, La Jolla, CA). All reported values are mean ± s.e.m. Shapiro-Wilk test was used to determine data normality. Data were analyzed using either Wilcoxon matched-pairs sign rank test for comparisons of non-parametric $P_{sleep}$ of animals before and after meals, Kruskal-Wallis test followed by Dunn's multiple comparisons for non-parametric comparisons of $\Delta P_{sleep}$, or the Mann-Whitney test when comparing two non-parametric groups, non-paired observations. Comparisons of meal size to given features were analyzed using Spearman rank correlation, due to non-parametric distribution, while binned data were analyzed using Pearson correlation.

## Acknowledgements

We thank M Southern for writing the image splitter plugin for ImageJ. We also thank A Wong, D Wilson, R Smith, A Chakraborty, K Reis Santos, and H Fuhrmann-Stroissnigg for comments on the manuscript. This work was supported by the NIH (R21DK092735), an Ellison Medical Foundation New Scholar in Aging award, and a Glenn Foundation for Medical Research Award for Research in Biological Mechanisms of Aging (WWJ).

## Additional information

### Funding

| Funder | Grant reference number | Author |
|---|---|---|
| National Institutes of Health | R21DK092735 | William W Ja |
| Ellison Medical Foundation | New Scholar in Aging award | William W Ja |
| Glenn Foundation for Medical Research | Medical Research Award for Research in Biological Mechanisms of Aging | William W Ja |

The funders had no role in study design, data collection and interpretation, or the decision to submit the work for publication.

### Author contributions

KRM, Conception and design, Acquisition of data, Analysis and interpretation of data, Drafting or revising the article; SAD, JPQ, Designed and carried out experiments, Acquisition of data; MEY, RH, Acquisition of data, Analysis and interpretation of data; JLW, Designed and carried out experiments, Acquisition of data, Contributed unpublished essential data or reagents; ACK, Designed and carried out experiments, Contributed unpublished essential data or reagents; KD-S, Developed the ARC, Conception and design; SMT, Conception and design, Acquisition of data, Analysis and interpretation of data; WWJ, Conception and design, Analysis and interpretation of data, Drafting or revising the article

### Author ORCIDs

William W Ja, http://orcid.org/0000-0002-4003-7356

## Additional files

### Supplementary files
• Supplementary file 1. Meal selection algorithm. Flowchart depicting the algorithm used for processing dye-reference mark pixel distance data to identify feeding events.

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
