## [Decision Letter]

[Editors’ note: a previous version of this study was rejected after peer review, but the authors submitted for reconsideration. The first decision letter after peer review is shown below.]

Thank you for submitting your work entitled "Postprandial sleep mechanics in *Drosophila*" for consideration by *eLife*. Your article has been reviewed by three peer reviewers, one of whom is a member of our Board of Reviewing Editors, and the evaluation has been overseen by a Senior Editor. Our decision has been reached after consultation between the reviewers. Based on these discussions and the individual reviews below, we regret to inform you that your work will not be considered further for publication in *eLife*.

The following individual involved in the review of your submission has agreed to reveal his identity: Paul Shaw (peer reviewer).

Although all three reviewers found the work of high quality and potentially interesting, the consensus opinion was that there was still a great deal of work required to support the conclusions of the manuscript. In particular, the studies of the role of leucokinin and LKR require additional experiments. Given the time it would take to perform these lengthy additional experiments, we felt it would be most fair to return the paper to you. If, in the future, you feel you can address these concerns, we would be open to receiving a new manuscript on the topic and would make every effort to return this to the same reviewers.

Reviewer #1:

This manuscript examines interactions between feeding and sleep in *Drosophila* and proposes that there is an increase in sleep post-feeding. The authors show decreased locomotion post-feeding, increased startle response, and that volume consumed and nutrient affects sleep probability. They further show that silencing a subset of LKR neurons decreased sleep whereas silencing LK neurons increased sleep, and interpreted this to suggest that LK has multiple roles in promoting and inhibiting sleep. Overall, there are intriguing findings in this manuscript; in particular the notion the feeding increases sleep is very exciting. The data, however, seem preliminary and more suitable for publication in a specialty journal. My main reservations are the following:

1) The role of LKR and LK is unclear. The nature of the lines used is not well-described. Do these lines really label a subset of all LKR neurons and all LK neurons? By what criteria, Ab staining?

2) The function of LK and LKR neurons is not clear. The observation that silencing LK neurons increased sleep and silencing LKR decreased sleep needs to be explained. Use of multiple RNAi lines to knock down LK and LKR, and use of multiple LK-Gal4 and LKR-Gal4 lines would be important to resolve this. Does RNAi against LK and LKR pan-neuronally increase sleep? Do different LKR and LK Gal4 lines show consistent behavioral phenotypes?

3) What is the evidence that the decreased locomotion is sleep? Flies may significantly increase grooming after feeding, leading to decreased movement overall. This, in principle, could influence startle and depend on ingested compound. High resolution monitoring of behavior would be important to rule out that feeding influences the probability of other behaviors.

Reviewer #2:

In this manuscript the authors study the impact of food intake on subsequent sleep. They report that the probability of sleeping is increased following meals which was correlated with the meal content. In addition, they report that silencing Leucokininreceptor expressing neurons reduces postprandial sleep. Similarly, silencing Leucokininneurons increase postprandial sleep.

All in all, I thought this was a well written and fun manuscript. The authors do a nice job of presenting the data and the results are solid. My main critique is that virtually all of the data is presented as a probability. While this may very well be the best way to present the data it leaves the reader without any way to assess the magnitude of the effect or compare it with changes in sleep reported in the literature in response to other manipulations.

With that in mind, it seems important to show some data that shows the changes in sleep in minutes and to describe other sleep parameters such as average sleep bout duration (if this data was included, I apologize but I couldn't find it). Small effects can have large consequences so there shouldn't be any hesitation about showing more data. Similarly, nobody has recorded sleep in flies maintained in a vertical environment. It would be great to see what the pattern of sleep looked like compared to the typical experiment.

To my knowledge yeast also contain lipids but there wasn't any attempt to determine how lipids changed postprandial sleep. This decision should be discussed.

In summary, I am very enthusiastic about this manuscript even though I am quite not sure if it will be functionally relevant for the animal.

Reviewer #3:

Postprandial sleepiness is a phenomenon that most can easily recognize, but has been difficult to objectively document. In this manuscript, Murphy et al. develop a novel assay combining the CAFE assay with video measurements of locomotor activity, in order to simultaneously assess feeding and sleep in *Drosophila*. Using this ARC assay, they find that larger meals, and protein and salt meals promote sleepiness, while sucrose meals do not. They also suggest that the Leuc/LKR pathway, previously implicated in regulating feeding in flies, modulates post-prandial sleepiness in response to protein intake.

Overall, the work is interesting and generally carefully done, and the ARC assay is likely to provide new insights into the dynamic relationship between feeding and sleep. However, the manuscript still needs additional work to justify its publication in a journal with broad impact and scope, like *eLife*.

1) The characterization of post-prandial sleepiness using the ARC assay has been carefully done. However, the novelty and impact of this work lies with the use of this assay to identify molecular or cellular mechanisms that mediate this process. To this point, the authors have significant work to do to make a convincing argument regarding the role of LKR and Leuc neurons in this post-prandial sleepiness.

a) The authors use a single LKR RNAi to knockdown LKR. QPCR data should be provided demonstrating effective knockdown of LKR and another independent RNAi strain (or alternative genetic approach) should be used to ensure specificity of phenotype.

b) The authors also use a single LKR-Gal4 line (R65C07) in their experiments. Has this driver been previously characterized, in terms of number of LKR+ cells vs LKR- cells, as well as precisely which LKR+ cells are present? As is the case with the RNAi line, an additional Gal4 line containing LKR cells (such as the original Lkr-Gal4-, AI-Anzi et al., 2010) would help to strengthen their claims. Similar issues exist with the authors' use of a single Leuc-Gal4 line (R20G03).

c) Given that they are examining adult behavior, it would be better to use conditional silencing approaches (e.g., Kir2.1, tubGal80ts) to bypass effects on development in assessing the role of LKR and Leuc neurons in post-prandial sleepiness. Also, does activation of LKR and Leuc neurons result in a postprandial phenotype opposite to that seen with silencing?

d) The authors make the unexpected finding that Leuc neurons (or at least a subset of them) seem to promote post-prandial arousal. In the absence of a better understanding of how the Leuc and LKR neurons interact to regulate sleep/wake, the Leuc data could be removed from the manuscript, as it does not add much.

e) Do the LKR or Leuc neurons show changes in activity with feeding? If they play a physiological role in post-prandial sleepiness, one would expect their activity levels to change with feeding (and with protein feeding in particular).

[Editors’ note: what now follows is the decision letter after the authors submitted for further consideration.]

Thank you for submitting your article "Postprandial sleep mechanics in *Drosophila*" for consideration by *eLife*. Your article has been reviewed by three peer reviewers, one of whom is a member of our Board of Reviewing Editors, and the evaluation has been overseen by a Senior Editor. The following individual involved in review of your submission has agreed to reveal his identity: Paul Shaw (Reviewer #2).

The reviewers have discussed the reviews with one another and the Reviewing Editor has drafted this decision to help you prepare a revised submission.

Summary:

This manuscript examines interactions between feeding and sleep in *Drosophila* and uncovers a role for LK in regulating postprandial sleep. In their resubmitted manuscript, Murphy et al. have performed a number of control experiments and focused their analysis of mechanisms underlying postprandial sleep to Leuc and LKR circuits. Since their last submission, Cavey et al. (2016) have published a description of Leuc/LKR neurons acting downstream of clock neurons to regulate locomotion and sleep, and also provide evidence for an inhibitory action of Leuc on LKR neurons, which helps to resolve the opposing effects of Leuc and LKR circuits on postprandial sleep previously seen by Murphy et al. Although the manuscript has considerably improved, questions remain about the Gal4 lines used and the roles of LK and LKR.

Essential revisions:

1) The Gal4 lines used for LK and LKR are Rubin transgenic lines that contain 3 Kb fragments of LK and LKR non-coding regions (these are not enhancer traps). Many times these fragments do not recapitulate endogenous gene expression. The Cavey study nicely validated the LKR line by showing that the neurons respond to LK. The LK line has not been validated and LK Ab staining with the Gal4 line used is required.

2) The authors removed the RNAi studies shown in the previous version, but these are important. Two RNAi lines for LK and LKR should be used to test the role of these genes in LK and LKR neurons (the Rubin Gal4 lines used in this study). If the LKR RNAi data conflicts with the model presented, then the model should be modified.

---

## [Author Response]

[Editors’ note: the author responses to the first round of peer review follow.]

[…] Although all three reviewers found the work of high quality and potentially interesting, the consensus opinion was that there was still a great deal of work required to support the conclusions of the manuscript. In particular, the studies of the role of leucokinin and LKR require additional experiments. Given the time it would take to perform these lengthy additional experiments, we felt it would be most fair to return the paper to you. If, in the future, you feel you can address these concerns, we would be open to receiving a new manuscript on the topic and would make every effort to return this to the same reviewers.

We thank the reviewers for their detailed comments. The manuscript has been substantially revised, as detailed in the point-by-point responses to reviewers below. The submission includes deeper characterization of the new behavioral paradigm so that results can be better interpreted. This characterization includes comparing different behavioral systems (ARC vs. DAM), demonstrating conserved postprandial sleep behavior in females, and a comprehensive examination of fly grooming behavior to verify the correlation between measured immobility and sleep.

We have also revised the manuscript to focus solely on the circuit regulation of postprandial sleep. We include new data for conditional silencing of Lk and Lkr neurons in adulthood, improved images for the neuronal drivers used and their interaction with one another, and the demonstration that Lk inhibitory signaling to Lkr neurons has a postprandial sleep effect that is both rhythmic and induced by feeding. This new data complements a recent publication (Cavey et al., Nature Neurosci. 2016) showing that Lk inhibits Lkr neurons in the fan-shaped body. We believe the new integrated system for *Drosophila* behavioral measurement, the description and validation of a new behavior, and the dissection of one aspect of its dietary and neuronal regulation are appropriate for consideration as an article in *eLife*.

*Reviewer #1:*

*[…] 1) The role of LKR and LK is unclear. The nature of the lines used is not well-described. Do these lines really label a subset of all LKR neurons and all LK neurons? By what criteria, Ab staining?*

Previous studies have characterized both LKR and LK cells in the Drosophila brain using antibody staining (De Haro et al., Cell Tissue Res. 2010 and Al-Anzi et al., Curr. Biol. 2010) which is recapitulated by the enhancer trap GAL4 lines used in this work. This is addressed in the text (“The expression pattern of R65C07 recapitulates that of Lkr neurons which were found to regulate meal size. […] Subsets of Lk neurons have been found in the lateral horn and are in close proximity to cell bodies of the Lkr neurons, suggesting neuromodulatory connection between these cells. This expression is recapitulated by R20G03, an enhancer trap line of Lk, which labels cell bodies in the lateral horn that coincide with Lkr neurons”). We have also re-focused the manuscript on the role of neuronal circuitry rather than Lk and Lkr within these circuits. The circuit-based section of the manuscript has been bolstered by substantial supportive data, and we believe the specific roles of LK and LKR are now better dissected in future work.

*2) The function of LK and LKR neurons is not clear. The observation that silencing LK neurons increased sleep and silencing LKR decreased sleep needs to be explained. Use of multiple RNAi lines to knock down LK and LKR, and use of multiple LK-Gal4 and LKR-Gal4 lines would be important to resolve this. Does RNAi against LK and LKR pan-neuronally increase sleep? Do different LKR and LK Gal4 lines show consistent behavioral phenotypes?*

We agree that the role of these neurons was previously unclear. The recent finding of an inhibitory interaction between Lk and Lkr neurons has clarified this opposing phenotype (Cavey et al. Nature Neurosci. 2016), which is now explained in the text. While we agree that a deep genetic interrogation of the behavior is also important, we think that doing so would be enough work for a separate manuscript and have in turn limited the current manuscript to a more thorough circuit-based analysis.

*3) What is the evidence that the decreased locomotion is sleep? Flies may significantly increase grooming after feeding, leading to decreased movement overall. This, in principle, could influence startle and depend on ingested compound. High resolution monitoring of behavior would be important to rule out that feeding influences the probability of other behaviors.*

We agree that this experiment is crucial to the validity of this study. We have addressed this by performing our typical experiment with simultaneous high resolution video capture. We manually scored grooming events and generated grooming ethograms for alignment with sleep data, allowing us to recalculate sleep while excluding immobile flies which were grooming. While grooming does account for ~7% of calculated sleep, we found that it does not artificially raise the calculated sleep after a meal any more than it raises the sleep prior (Figure 2, and Figure 2—figure supplement 3).

*Reviewer #2:*

*In this manuscript the authors study the impact of food intake on subsequent sleep. They report that the probability of sleeping is increased following meals which was correlated with the meal content. In addition, they report that silencing Leucokinin receptor expressing neurons reduces postprandial sleep. Similarly, silencing Leucokinin neurons increase postprandial sleep.*

*All in all, I thought this was a well written and fun manuscript. The authors do a nice job of presenting the data and the results are solid. My main critique is that virtually all of the data is presented as a probability. While this may very well be the best way to present the data it leaves the reader without any way to assess the magnitude of the effect or compare it with changes in sleep reported in the literature in response to other manipulations.*

*With that in mind, it seems important to show some data that shows the changes in sleep in minutes and to describe other sleep parameters such as average sleep bout duration (if this data was included, I apologize but I couldn't find it). Small effects can have large consequences so there shouldn't be any hesitation about showing more data.*

We thank the reviewer for the positive remarks and suggestions. While we recognize that probability is a non-traditional method for presenting sleep, this behavior occurs on a much shorter time scale than other data sets which are better shown in the typical 30 minute bins. Since there are no sleep-related behaviors, to our knowledge, which use 1-minute time bins we decided it was most intuitive to think of the changes in sleep as probability rather than seconds per minute. However, we agree that experts in the field should be given a relatable scale for comparison with their own data, and so a seconds per minute scale was added to the first sleep probability graphs shown (Figure 2) as well as the graph of sleep probability related meal size (Figure 3). We also recognize the utility in further examining the sleep architecture surrounding a meal. However, since sleep duration in the time leading up to a meal is restricted by the fact that the animal cannot sleep while eating, we had originally excluded any examinations of sleep architecture. Since sleep initiation is not restricted we have included a supplemental figure showing the probability of sleep initiation as well as the sleep frequency before and after meals (Figure 2—figure supplement 1).

*Similarly, nobody has recorded sleep in flies maintained in a vertical environment. It would be great to see what the pattern of sleep looked like compared to the typical experiment.*

Comparison of sleep in a vertical versus horizontal chamber is an excellent suggestion and should serve to inform the *Drosophila* community about the influence of environmental orientation on fly sleep. We have now included a supplementary figure comparing sleep data from animals in an ARC or the traditional DAM system (Figure 1—figure supplement 1). Although the type of data (video versus IR beam breaks) and calculation of sleep is different in the two systems, sleep patterns were similar and we believe the comparison makes our findings more relatable to those using the classical system.

*To my knowledge yeast also contain lipids but there wasn't any attempt to determine how lipids changed postprandial sleep. This decision should be discussed.*

Because the food used in the experiments is water based, and because oil is used to mark the meniscus for high resolution feeding measurement, lipids have been extremely difficult to measure in the ARC (and with the capillary feeder or CAFE assay in general). However, we are beginning to test various methods for measuring lipid consumption in high resolution and have added a small section discussing the potential interest in these findings (“Lipids are also present in many fly food sources and future examination of their influence on postprandial sleep might be facilitated by adding fat solubilizers to the liquid food used in the ARC.”)

*In summary, I am very enthusiastic about this manuscript even though I am quite not sure if it will be functionally relevant for the animal.*

We thank the reviewer for these comments. We believe the strong conservation of postprandial somnolence in different species (at least anecdotally) suggests an important function that will be revealed in future work.

*Reviewer #3:*

*[…] 1) The characterization of post-prandial sleepiness using the ARC assay has been carefully done. However, the novelty and impact of this work lies with the use of this assay to identify molecular or cellular mechanisms that mediate this process. To this point, the authors have significant work to do to make a convincing argument regarding the role of LKR and Leuc neurons in this post-prandial sleepiness.*

We agree and have refocused the manuscript on additional circuitry data.

*a) The authors use a single LKR RNAi to knockdown LKR. QPCR data should be provided demonstrating effective knockdown of LKR and another independent RNAi strain (or alternative genetic approach) should be used to ensure specificity of phenotype.*

As mentioned above, the manuscript has been refocused on neuronal circuitry rather than genetic mechanisms. Hence, the RNAi data have been removed.

*b) The authors also use a single LKR-Gal4 line (R65C07) in their experiments. Has this driver been previously characterized, in terms of number of LKR+ cells vs LKR- cells, as well as precisely which LKR+ cells are present? As is the case with the RNAi line, an additional Gal4 line containing LKR cells (such as the original Lkr-Gal4, AI-Anzi et al., 2010) would help to strengthen their claims. Similar issues exist with the authors' use of a single Leuc-Gal4 line (R20G03).*

To our knowledge, the only characterizations of Lk and LKR cells have been done by Al-Anzi et al. and de Haro et al., which very closely label the same cells. However, enhancer trap lines tend to show more variability in cell labeling between animals so it would be difficult to verify how many Lkr– cells are labeled within an experiment. We have now included higher resolution images of both the Lkr and Lk cells labeled by the R65C07 and R20G03 cells (Figure 5 and Figure 6), respectively, which can be visually compared to the drivers used by Al-Anzi et al. In addition, Cavey et.al were able to demonstrate phenotypes using these same enhancer trap lines which supported their genetic findings.

*c) Given that they are examining adult behavior, it would be better to use conditional silencing approaches (e.g., Kir2.1, tubGal80ts) to bypass effects on development in assessing the role of LKR and Leuc neurons in post-prandial sleepiness. Also, does activation of LKR and Leuc neurons result in a postprandial phenotype opposite to that seen with silencing?*

This is an excellent suggestion to support the neuronal findings and we now include studies with tubGal80ts to conditionally silence the Lkr or Lk neurons while observing postprandial sleep (Figure 5Figure 7). While we have attempted activation of these neurons using TrpA, the results have not been strong enough for us to make any conclusions on the data. It is notable that both the Lk and Lkr cells seem to have very low basal activity and that TrpA activation may quickly hyperpolarize the cells or might have non-physiologically relevant activation.

*d) The authors make the unexpected finding that Leuc neurons (or at least a subset of them) seem to promote post-prandial arousal. In the absence of a better understanding of how the Leuc and LKR neurons interact to regulate sleep/wake, the Leuc data could be removed from the manuscript, as it does not add much.*

During the revision of this paper, Cavey et al. showed that Lkr neurons respond to carbachol, a cholinergic agonist, and that incubating Lkr neurons with Leucokinin shunted this response. While this strongly suggested that the lateral horn Lk neurons inhibit the connecting Lkr neurons, there was very little evidence of this on a behavioral level. Our data shows a clear opposing phenotype when silencing Lkr versus Lk neurons, shedding new light on their findings. Cavey et al. also showed that Lk signaling peaked at ZT 10-14 using Gcamp6 in explanted brains. We were able to expand our analysis to show that the inhibitory effect of Leucokinin was greatest during this time period. We think our findings are strongly supported by and complementary to the evidence from Cavey et al. of Lk-Lkr inhibition.

*e) Do the LKR or Leuc neurons show changes in activity with feeding? If they play a physiological role in post-prandial sleepiness, one would expect their activity levels to change with feeding (and with protein feeding in particular).*

We do observe a trend for increased activity; however the difference is not consistently significant across experiments. We have highlighted this in the results (“This manipulation eliminated any rise in postprandial sleep and instead caused slight arousal (Figure 5). There was no significant change in postprandial activity (Δ_motion_ for 20 min surrounding meal, *Kir^2.1^*/ *w^CS^*, -0.006 ± 0.003; *Lkr^Gal4^*/ *w^CS^*, -0.001 ± 0.002; *Kir^2.1^*/ *Lkr^Gal4^* 0.004 ± 0.004; p = 0.28, Kruskal-Wallis test followed by Dunn’s Multiple Comparison).”).

[Editors' note: the author responses to the re-review follow.]

*[…] Essential revisions:*

*1) The Gal4 lines used for LK and LKR are Rubin transgenic lines that contain 3 Kb fragments of LK and LKR non-coding regions (these are not enhancer traps). Many times these fragments do not recapitulate endogenous gene expression. The Cavey study nicely validated the LKR line by showing that the neurons respond to LK. The LK line has not been validated and LK Ab staining with the Gal4 line used is required.*

R20G03 labels several cells in proximity and with similar morphology to Lk neurons from previous studies. However, anti-Lk immunostaining did not show Lk co-localization in any of the cells labeled by this driver (Figure 7—figure supplement 1). Additionally, promoting Lk RNAi in the cells had no effect on postprandial sleep, in agreement with the lack of Lk expression in *R20G03*-labeled cells.

We have also now tested the previously validated Lk-Gal4, which we confirmed does label cells expressing Lk (Figure 6). TRPA stimulation of these cells, or Lk knockdown by RNAi, are consistent with the idea that they are inhibitory, in agreement with Cavey et al. Thus, the revised manuscript now describes two populations of cells labeled by R20G03 or Lk-Gal4 that are independent or dependent on leucokinin, respectively. Our proposed model suggests that these cells function through modulating Lkr neuronal activity, but further studies are needed to confirm these hypotheses.

*2) The authors removed the RNAi studies shown in the previous version, but these are important. Two RNAi lines for LK and LKR should be used to test the role of these genes in LK and LKR neurons (the Rubin Gal4 lines used in this study). If the LKR RNAi data conflicts with the model presented, then the model should be modified.*

We have now demonstrated the role of LK in Lk-Gal4 labeled cells using two independent RNAi lines. Knockdown of Lk using either line increased postprandial sleep, consistent with the putative Lk-Lkr inhibitory action found by Cavey et al. As an additional control, we have also shown that Lk RNAi does not have an effect when driven in R20G03 cells (Figure 7—figure supplement 1). We were ultimately only able to obtain one reliable RNAi line for Lkr. While results using this line were robust, we have left this series of experiments for future work where we will generate additional independent RNAi lines to validate the observed effects. The proposed model (Figure 7) has been modified and the text has been revised to reflect the additional unknown genetic interactions and neuronal connections that must describe postprandial sleep.